# Learning to Teach with Dynamic Loss Functions

[1]**Lijun Wu**[†,∗] [2]**Fei Tian**,[†] [2]**Yingce Xia,** [3]**Yang Fan**[⋆],
[2]**Tao Qin,** [1]**Jianhuang Lai,** [2]**Tie-Yan Liu**
[1]Sun Yat-sen University, Guangzhou, China    [2]Microsoft Research, Beijing, China
[3]University of Science and Technology of China, Hefei, China
[1]wulijun3@mail2.sysu.edu.cn, stsljh@mail.sysu.edu.cn
[2]{fetia, yingce.xia, taoqin, tie-yan.liu}@microsoft.com, [3]fyabc@mail.ustc.edu.cn

## Abstract

Teaching is critical to human society: it is with teaching that prospective students are educated and human civilization can be inherited and advanced. A good teacher not only provides his/her students with qualified teaching materials (e.g., textbooks), but also sets up appropriate learning objectives (e.g., course projects and exams) considering different situations of a student. When it comes to artificial intelligence, treating machine learning models as students, the loss functions that are optimized act as perfect counterparts of the learning objective set by the teacher. In this work, we explore the possibility of imitating human teaching behaviors by dynamically and automatically outputting appropriate loss functions to train machine learning models. Different from typical learning settings in which the loss function of a machine learning model is predefined and fixed, in our framework, the loss function of a machine learning model (we call it student) is defined by another machine learning model (we call it teacher). The ultimate goal of teacher model is cultivating the student to have better performance measured on development dataset. Towards that end, similar to human teaching, the teacher, a parametric model, dynamically outputs different loss functions that will be used and optimized by its student model at different training stages. We develop an efficient learning method for the teacher model that makes gradient based optimization possible, exempt of the ineffective solutions such as policy optimization. We name our method as "learning to teach with dynamic loss functions" (L2T-DLF for short). Extensive experiments on real world tasks including image classification and neural machine translation demonstrate that our method significantly improves the quality of various student models.

## 1 Introduction

Teaching, which aims to help students learn new knowledge or skills effectively and efficiently, is important to advance modern human civilization. In human society, the rapid growth of qualified students not only relies on their intrinsic learning capability, but also, even more importantly, relies on the substantial guidance from their teachers. The duties of teachers cover a wide spectrum: defining the scope of learning (e.g., the knowledge and skills that we expect students to demonstrate by the end of a course), choosing appropriate instructional materials (e.g., textbooks), and assessing the progress of students (e.g., through course projects or exams). Effective teaching involves progressively and dynamically refining the teaching strategy based on reflection and feedback from students.

Recently, the concept of teaching has been introduced into artificial intelligence (AI), so as to improve the learning process of a machine learning model. Currently, teaching in AI mainly focuses on

---

∗The work was done when the first and fourth authors were interns at Microsoft Research Asia.
†The first two authors contribute equally to this work.

training data selection. For example, *machine teaching* [56, 34, 35] aims at identifying the smallest training data that is capable of producing the optimal learner models. The very recent work, *learning to teach* (L2T for short) [13], demonstrates how to automatically design teacher models for better machine learning process. While conceptually L2T can cover different aspects of teaching in AI, [13] only studies the problem of training data teaching.

In this work, inspired from learning to teach, we study loss function teaching in a formal and concrete manner for the first time. The main motivation of our work is a natural observation on the analogy between loss functions in machine learning and exams in educating human students: appropriate exams reflect the progress of students and urge them to make improvements accordingly, while loss values outputted by the loss function evaluate the performance of current machine learning model and set the optimization direction for the model parameters.

In our loss function teaching framework, a teacher model plays the role of outputting loss functions for the student model (i.e., the daily machine learning model to solve a task) to minimize. Inspired from human teaching, we design the teacher model according to the following principles. First, similar to the different difficulty levels of exams with respect to the progress of student in human education, the loss function set by the teacher model should be *dynamic*, i.e., the loss functions should be adaptive to different phases of the training process of the student model. To achieve this, we require our teacher model to take the status of student model into consideration in setting the loss functions, and to dynamically change the loss functions with respect to the growth of the student model. Such process is shown in Fig. 1. Second, the teacher model should be able to make self-improvement, just as a human teacher can accumulate more knowledge and improve his/her teaching skills through more teaching practices. To achieve that, we assume the loss function takes the form of neural network whose coefficients are determined via a parametric teacher model, which is also a neural network. The parameters of the teacher model can be automatically optimized in the teaching process. Through optimization, the teacher keeps improving its teaching model and consequently the quality of loss functions it outputs. We name our method as *learning to teach with dynamic loss functions* (L2T-DLF).

The eventual goal of the teacher model is that its output can serve as the loss function of the student model to maximize the long-term performance of the student, measured via a task-specific objective such as 0-1 accuracy in classification and BLEU score in sequence prediction [41], on a stand-alone development dataset. Learning a good teaching model is not trivial, since on the one hand the task-specific objective is usually non-smooth w.r.t. student model outputs, and on the other hand the final evaluation of the student model is incurred on the dev set, disjoint with the training dataset where the teaching process actually happens. We design an efficient gradient based optimization algorithm to optimize teacher models. Specifically, to tackle the first challenge, we smooth the task-specific measure to its expected version where the expectation is taken on the direct output of student model. To address the second challenge, inspired by Reverse-Mode Differentiation (RMD) [6, 7, 38], through reversing the stochastic gradient descent training process of the student model, we obtain derivatives of the parameters of the teacher model via chaining backwards the error signals incurred on the development dataset .

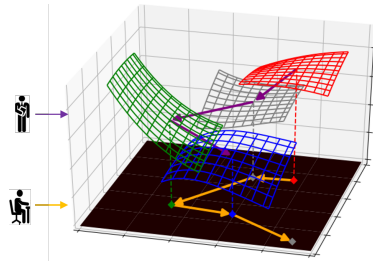

Figure 1: The student model is trained via minimizing the dynamic loss functions taught by the teacher model (yellow curve). The bottom black plane represents the parameter space of student model and the four colored mesh surfaces denote different loss functions outputted via a teacher model at different phases of student model training.

We demonstrate the effectiveness of L2T-DLF on various real-world tasks including image classification and neural machine translation with different student models such as multi-layer perception networks, convolutional neural networks and sequence-to-sequence models with attention. The improvements clearly demonstrate the effectiveness of the new loss function learnt by L2T-DLF.

## 2 Related Work

The study of teaching for AI, inspired by human teaching process, has a long history [1, 17]. The most recent efforts of teaching mainly focus on the level of training data selection. For example, the *machine teaching* [56, 34, 35] literature targets at building the smallest training set to obtain a pre-given optimal student model. A teaching strategy is designed in [18, 19] to iteratively select unlabeled data to label within the context of multi label propagation, in a similar manner with curriculum learning [8, 27]. Furthermore there are research on *pedagogical teaching* inspired from cognitive science [44, 23, 39] in which a teacher module is responsible for providing informative examples to the learner for the sake of understanding a concept rapidly.

The recent work *learning to teach* (L2T) [13] offers a more comprehensive view of teaching for AI, including training data teaching, loss function teaching and hypothesis space teaching. Furthermore, L2T breaks the strong assumption towards the existence of an optimal off-the-shelf student model adopted by previous *machine teaching* literature [56, 35]. Our work belongs to the general framework of L2T, with a particular focus on a thorough landscape of loss function teaching, including the detailed problem setup and efficient solution for dynamically setting loss functions for training machine learning models.

Our work, and the more general L2T, leverages automatic techniques to bypass human prior knowledge as much as possible, which is in line with the principles of *learning to learn* and *meta learning* [43, 50, 2, 57, 37, 29, 10, 14]. What makes our work different with others, from the technical point of view, is that: 1) we leverage gradient based optimization method rather than reinforcement learning [57, 13]; 2) we need to handle the difficulty when the error information cannot be directly back propagated from the loss function, since we aim at discovering the best loss function for the machine learning models. We design an algorithm based on Reverse-Mode Differentiation (RMD) [7, 38, 15] to tackle such a difficulty.

Specially designed loss functions play important roles in boosting the performances of real-world tasks, either by approximating the non-smooth task-specific objective such as 0-1 accuracy in classification [40], NDCG in ranking [49], BLEU in machine translation [45, 3] and MAP in object detection [22, 46], or easing the optimization process of the student model such as overcoming the difficulty brought by data imbalance [30, 32] and numerous local optima [20]. L2T-DLF differs from prior works in that: 1) the loss functions are automatically learned, covering a large space and without the demand of heuristic understanding for task specific objective and optimization process; 2) the loss function dynamically evolves during the training process, leading to a more coherent interaction between loss and student model.

## 3 Model

In this section, we introduce the details of L2T-DLF, including the student model and the teacher model, as well as the training strategy for optimizing the teacher model.

### 3.1 Student Model

For a task of interest, we denote its input space and output space respectively as $\mathcal{X}$ and $\mathcal{Y}$. The student model for this task is then denoted as $f_\omega : \mathcal{X} \to \mathcal{Y}$, with $\omega$ as its weight parameters. The training of student model $f_\omega$ is an optimization process that discovers a good weight parameter $\omega^*$ within a hypothesis space $\Omega$, by minimizing a loss function $l$ on the training data $D_{train}$ containing $M$ data points $D_{train} = \{(x_i, y_i)\}_{i=1}^{M}$. Specifically $\omega^*$ is obtained via solving $\min_{\omega \in \Omega} \sum_{(x,y) \in D_{train}} l(f_\omega(x), y)$. For the convenience of description, we define a new notation $L(f_\omega, D) = \sum_{(x,y) \in D} l(f_\omega(x), y)$ where $D$ is a dataset and will simultaneously name $L$ as loss function when the context is clear. The learnt student model $f_{\omega^*}$ is then evaluated on a test data set $D_{test} = \{(x_i, y_i)\}_{i=1}^{N}$ to obtain a score $\mathcal{M}(f_{\omega^*}, D_{test}) = \sum_{(x,y) \in D_{test}} m(f_{\omega^*}(x), y)$, as its performance. Here the task specific objective $m(y_1, y_2)$ measures the similarity between two output candidates $y_1$ and $y_2$.

The loss function $l(\hat{y}, y)$, taking the model prediction $\hat{y} = f_\omega(x)$ and ground-truth $y$ as inputs, acts as the surrogate of $m$ to evaluate the student model $f_\omega$ during its training process, just as the exams in real-world human teaching. We assume $l(\hat{y}, y)$ is a neural network with some coefficients $\Phi$, denoted as $l_\Phi(\hat{y}, y)$. It can be a simple linear model, or a deep neural network (some concrete examples

are provided in section 4.1 and section 4.2). With such a loss function $l_\Phi(\hat{y}, y)$ (and the induced notation $L_\Phi$), the student model gets sequentially updated via minimizing the output value of $l_\Phi$ by, for example, stochastic gradient descent (SGD): $\omega_{t+1} = \omega_t - \eta_t \frac{\partial L_\Phi(f_{\omega_t}, D^t_{train})}{\partial \omega_t}, t = \{1, 2, \cdots, T\}$, where $D^t_{train} \subseteq D_{train}$, $\omega_t$ and $\eta_t$ is respectively the mini-batch training data, student model weight parameter and learning rate at $t$-th timestep. For ease of statement we simply set $\omega^* = \omega_T$.

## 3.2 Teacher Model

A teacher model is responsible for setting the proper loss function $l$ to the student model by outputting appropriate loss function coefficients $\Phi$. To cater for different status of student model training, we ask the teacher model to output different loss functions $l^t$ at each training step $t$. To achieve that, the status of a student model is represented by a state vector $s_t$ at timestep $t$, which contains for example the current training/dev accuracy and iteration number. The teacher model, denoted as $\mu$, then takes $s_t$ as inputs to compute the coefficients of loss function $\Phi_t$ at $t$-th timestep as $\Phi_t = \mu_\theta(s_t)$, where $\theta$ is the parameters of the teacher model. We further provide some examples of $\mu_\theta$ in section 4.1 and section 4.2. The actual loss function for student model is then $l^t = l_{\Phi_t}$. The learning process of student model then switches to:

$$\omega_{t+1} = \omega_t - \eta_t \frac{\partial L_{\Phi_t}(f_{\omega_t}, D^t_{train})}{\partial \omega_t} = \omega_t - \eta_t \frac{\partial L_{\mu_\theta(s_t)}(f_{\omega_t}, D^t_{train})}{\partial \omega_t}. \tag{1}$$

Such a sequential procedure of obtaining $f_{\omega^*}$ (i.e., $f_{\omega_T}$) is the learning process of the student model with training data $D_{train}$ and loss function provided via the teacher model $\mu_\theta$, and we use an abstract operator $\mathcal{F}$ to denote it: $f_{\omega^*} = \mathcal{F}(D_{train}, \mu_\theta)$.

Just as the training and testing setup in typical machine learning scenarios, the teacher model here similarly follows the two phases setup. Specifically, in the training process of teacher model, similar to qualified human teachers are good at improving the quality of exams, the teacher model in L2T-DLF refines the loss function it sets up via optimizing its own $\theta$. The ultimate goal of teacher model is to maximize the performance of induced student model on a stand-alone development dataset $D_{dev}$:

$$\max_\theta \mathcal{M}(f_{\omega^*}, D_{dev}) = \max_\theta \mathcal{M}(\mathcal{F}(D_{train}, \mu_\theta), D_{dev}). \tag{2}$$

We introduce the detailed training process (i.e., how to efficiently optimize Eqn. (2)) in section 3.3. In the testing process of the teacher model, $\theta$ is fixed and the student model $f_\omega$ gets updated with the guidance of teacher model $\mu_\theta$, as specified in Eqn. (1).

## 3.3 Training Process of Teacher Model

There are two challenges to optimize teacher model: 1) the evaluation measure $m$ is typically non-smooth and non-differentiable w.r.t. the parameters of student model; 2) the error is incurred on dev set while the teacher model plays effect in training phase.

We use continuous relaxation of $m$ to tackle the first challenge. The main idea is to inject randomness into $m$ to form an approximated version $\tilde{m}$, where the randomness comes from the student model [49]. Thanks to the fact that quite a few student models output probabilistic distributions on $\mathcal{Y}$, the randomness naturally comes from the direct outputs of $f_\omega$. Specifically, to approximate the performance of $f_\omega$ on a test data sample $(x, y)$, we have $\tilde{m}(f_\omega(x), y) = \sum_{y^* \in \mathcal{Y}} m(y^*, y) p_\omega(y^* | x)$, where $p_\omega(y^* | x)$ is the probability of predicting $y^*$ given $x$ using $f_\omega$. The gradient of $\omega$ is then easy to obtain via $\frac{\partial \tilde{m}(f_\omega(x), y)}{\partial \omega} = \sum_{y^* \in \mathcal{Y}} m(y^*, y) \frac{\partial p_\omega(y^* | x)}{\partial \omega}$. We further introduce a new notation $\tilde{\mathcal{M}}(f_\omega, D_{dev}) = \sum_{(x,y) \in D_{dev}} \tilde{m}(f_\omega(x), y)$ which approximates the objective of the teacher model $\mathcal{M}(f_{\omega_T}, D_{dev})$.

We use Reverse-Mode Differentiation (RMD) [6, 7, 38] to fill in the gap between training data and development data. To better show the RMD process, we can view the sequential process in Eqn. (1) as a special feed-forward process of a deep neural network where each $t$ corresponds to one layer, and RMD corresponds to the backpropagation process looping the SGD process backwards from $T$ to 1. Specifically denote $d\theta$ as the gradient of $\tilde{M}(f_{\omega_T}, D_{dev})$ w.r.t. the teacher model parameters $\theta$, which has initial value $d\theta = 0$. On the dev dataset $D_{dev}$, the gradient of $\tilde{\mathcal{M}}(f_\omega, D_{dev})$ w.r.t. the

parameter of student model $\omega_T$ is calculated as

$$d\omega_T = \frac{\partial \tilde{\mathcal{M}}(f_{\omega_T}, D_{dev})}{\partial \omega_T} = \sum_{(x,y) \in D_{dev}} \frac{\partial \tilde{m}(f_{\omega_T}(x), y)}{\partial \omega_T}. \tag{3}$$

Then looping backwards from $T$ and corresponding to Eqn. (1), at each step $t = \{T-1, \cdots, 1\}$ we have

$$d\omega_t = \frac{\partial \tilde{\mathcal{M}}(f_{\omega_t}, D_{dev})}{\partial \omega_t} = d\omega_{t+1} - \eta_t \frac{\partial^2 L_{\mu_\theta(s_t)}(f_{\omega_t}, D_{train}^t)}{\partial \omega_t^2} d\omega_{t+1}. \tag{4}$$

At the same time, the gradient of $\tilde{\mathcal{M}}$ w.r.t. $\theta$ is accumulated at this time step as:

$$d\theta = d\theta - \eta_t \frac{\partial^2 L_{\mu_\theta(s_t)}(f_{\omega_t}, D_{train}^t)}{\partial \theta \partial \omega_t} d\omega_{t+1}. \tag{5}$$

We leave the detailed derivations for Eqn. (4) and (5) to Appendix. Furthermore it is worth-noting that the computing of $d\omega_t$ and $d\theta$ involves hessian vector product, which can be effectively computed via $\frac{\partial^2 g}{\partial x \partial y} v = \partial(\frac{\partial g}{\partial y} v)/\partial x$, without explicitly calculating the Hessian matrix. Reverting backwards from $t = T$ to $t = 1$, we obtain $d\theta$ and then $\theta$ is updated using any gradient based optimization algorithm such as momentum SGD, forming one step optimization for $\theta$ which we call *teacher optimization step*. By iterating teacher optimization steps we obtain the final teacher model. The details are listed in Algorithm 1.

---

**Algorithm 1** Training Teacher Model $\mu_\theta$

---

**Input**: Continuous relaxation $\tilde{m}$. Initial value of $\theta$.
**while** Teacher model parameter $\theta$ not converged **do**        ▷ One *teacher optimization step*
    Randomly initialize student model parameter $\omega_0$.
    **for** each time step $t = 0, \cdots, T-1$ **do**        ▷ Teach student model
        Conduct student model training step via Eqn. (1).
    **end for**
    $d\theta = 0$. Compute $d\omega_T$ via Eqn. (3).
    **for** each time step $t = T-1, \cdots, 0$ **do**        ▷ Reversely calculating the gradient $d\theta$
        Update $d\theta$ as Eqn. (5).
        Compute $d\omega_t$ as Eqn. (4).
    **end for**
    Update $\theta$ using $d\theta$ via gradient based optimization algorithm.
**end while**
**Output**: the final teacher model $\mu_\theta$.

---

### 3.4 Discussion

Another possible way to conduct teacher model optimization is through deep reinforcement learning. By treating the teacher model as a policy outputting continuous action (i.e., the loss function), one can leverage continuous control algorithm such as DDPG [31] to optimize teacher model. However, reinforcement learning algorithms, including Q-learning based ones such as DDPG are sample inefficient, probably requiring huge amount of sampled trajectories to approximate the reward using a critic network. Considering the training of student model is typically costly, we resort to gradient based optimization algorithms instead.

Furthermore, there are similarity between L2T-DLF and actor-critic (AC) method [5, 48] in reinforcement learning (RL), in which a critic (corresponding to the parametric loss function) guides the optimization of an actor (corresponding to the student model). Apart from the difference within application domain (supervised learning versus RL), there are differences between the design principle of L2T-DLF and AC. For AC, by treating student model as actor, the student model output (e.g., $f_{\omega_t}(x_t)$) is essentially the action at timestep $t$, fed into the critic to output an approximation to the future reward (e.g., dev set accuracy). This is typically difficult since: 1) the student model output (i.e., the action) at a particular step $t$ is weakly related with the final dev performance. Therefore optimizing its action with the guidance from critic network is largely meaningless; 2) the approximation to the future

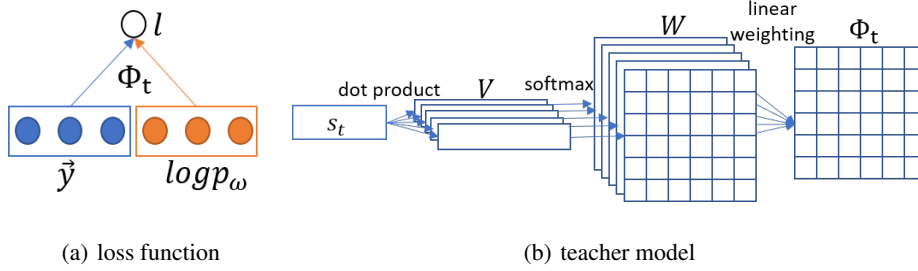

|     |     |
| --- | --- |
| (a) loss function | (b) teacher model |

Figure 2: Left: the bilinear neural network specifying the loss function $l_{\Phi_t}(p_\omega, y) = -\sigma(\vec{y}' \Phi_t \log p_w)$. Right: the teacher model outputting $\Phi_t$ via attention mechanism: $\Phi_t = \mu_\theta(s_t) = W softmax(V s_t)$.

reward is hard given the performance measure is highly non-smooth. As a comparison, L2T-DLF is more general in that at each timestep: 1) the teacher model considers the overall status of the student model for the sake of optimizing its parameters, rather than the instant action (i.e., the direct output); 2) the teacher model outputs a loss function with the goal of maximizing, but not approximating the future reward. In that sense, L2T-DLF is more appropriate to real world applications.

## 4 Experiments

We conduct comprehensive empirical verifications of the proposed L2T-DLF, in automatically discovering the most appropriate loss functions for student model training. The tasks in our experiments come from two domains: image classification, and neural machine translation.

### 4.1 Image Classification

The evaluation measure $m$ here is the 0-1 accuracy: $m(y_1, y_2) = \mathbb{1}_{y_1 = y_2}$ where $\mathbb{1}$ is the 0-1 indicator function. The student model $f_\omega$ can be a logistic classifier specifying a softmax distribution $p_\omega(y|x) = \exp\left(w'_y x + b_y\right) / \sum_{y^* \in \mathcal{Y}} \exp\left(w'_{y^*} x + b_{y^*}\right)$ with $\omega = \{w_{y^*}, b_{y^*}\}_{y^* \in \mathcal{Y}}$. The class label is predicted as $\hat{y} = \arg\max_{y^* \in \mathcal{Y}} p_\omega(y^*|x)$ given input data $x$. Instead of imposing loss on $\hat{y}$ and ground-truth $y$, for the sake of efficient optimization $l$ typically takes the direct model output $p_\omega$ and $y$ as inputs. For example, the most widely adopted loss function $l$ is cross-entropy loss $l(p_\omega, y) = -\log p_\omega(y|x)$, which could be re-written in vector form $l(p_\omega, y) = -\vec{y}' \log p_\omega$, where $\vec{y} \in \{0, 1\}^{|\mathcal{Y}|}$ is a one-hot representation of the true label $y$, i.e., $\vec{y}_j = \mathbb{1}_{j=y}, \forall j \in \mathcal{Y}$, $\vec{y}'$ is the transpose of $\vec{y}$ and $p_w \in \mathcal{R}^{|\mathcal{Y}|}$ is the probabilities for each class outputted via $f_\omega$.

Generalizing the cross entropy loss, we set the loss function coefficients $\Phi$ as a matrix interacting between $\log p_w$ and $\vec{y}$, which switches loss function at $t$-th timestep into $l_{\Phi_t}(p_\omega, y) = -\sigma(\vec{y}' \Phi_t \log p_w)$, $\Phi_t \in \mathcal{R}^{|\mathcal{Y}| \times |\mathcal{Y}|}$, as is shown in Fig. 2(a). $\sigma$ is the sigmoid function. The teacher model $\mu_\theta$ here is then responsible for setting $\Phi_t$ according to the state feature vector of student model $s_t$: $\Phi_t = \mu_\theta(s_t)$. One possible form of the teacher model is a neural network with attention mechanism (shown in Fig. 2(b)): $\Phi_t = \mu_\theta(s_t) = W softmax(V s_t)$, where $W \in \mathcal{R}^{|\mathcal{Y}| \times |\mathcal{Y}| \times N}, V \in \mathcal{R}^{N \times |s_t|}$ constitute the teacher model parameter set $\theta$, $N = 10$ is the number of keys in attention mechanism. The state vector $s_t$ is a 13 dimensional vector composing of 1) the current iteration number $t$; 2) current training accuracy of $f_\omega$; 3) current dev accuracy of $f_\omega$; 4) current precision of $f_\omega$ for the 10 classes on the dev set, all normalized into $[0, 1]$.

We choose three widely adopted datasets: the MNIST, CIFAR-10 and CIFAR-100 datasets. For the sake of showing the robustness of L2T-DLF, the student models we choose cover a wide range, including multi-layer perceptron (MLP), plain convolutional neural network (CNN) following LeNet architecture [28], and advanced CNN architecture including ResNet [21], Wide-ResNet [55] and DenseNet [24]. For all the student models, we use momentum stochastic gradient descent to perform training. In Appendix we describe the network structures of student models.

The different loss functions we compare include: 1) Cross entropy loss $L_{ce}(p_\omega(x), y) = -\log p_\omega(y|x)$, which is the most widely adopted loss function to train neural network model;

Table 1: The recognition results (error rate %) on MNIST dataset.

| Student Model/ Loss | Cross Entropy [11] | Smooth [40] | Large-Margin Softmax [36] | L2T-DLF |
|---|---|---|---|---|
| *MLP* | 1.94 | 1.89 | 1.83 | **1.69** |
| *LeNet* | 0.98 | 0.94 | 0.88 | **0.77** |

Table 2: The recognition results (error rate %) on CIFAR-10 (C10) and CIFAR-100 (C100) dataset

| Student Model/ Loss | Cross Entropy [11] | Smooth [40] | Large-Margin Softmax [36] | L2T-DLF |
|---|---|---|---|---|
| | C10/C100 | C10/C100 | C10/C100 | C10/C100 |
| *ResNet-8* | 12.45/39.79 | 12.08/39.52 | 11.34/38.93 | **10.82/38.27** |
| *ResNet-20* | 8.75/32.33 | 8.53/32.01 | 8.02/31.65 | **7.63/30.97** |
| *ResNet-32* | 7.51/30.38 | 7.42/30.12 | 7.01/29.56 | **6.95/29.25** |
| *WRN* | 3.80/- | 3.81/- | 3.69/- | **3.42/-** |
| *DenseNet-BC* | 3.54/- | 3.48/- | 3.37/- | **3.08/-** |

2) The smooth 0-1 loss proposed in [40]. It optimizes a smooth version of 0-1 accuracy in binary classification. We extend it to handle multi-class case by modifying the loss function as $L_{smooth}(p_\omega(x), y) = -\log \sigma(K(\log p_\omega(y|x) - \max_{y^* \neq y} \log p_\omega(y^*|x)))$. It is not difficult to observe when $K \to +\infty$, $-L_{smooth}$ exactly matches the 0-1 accuracy. We choose the value of $K$ to be 50 according to the performance on dev set; 3) The large-margin softmax loss in [36] denoted as $L_{lm}$, which aims to enhance discrimination between different classes via maximizing the margin induced by the angle between $x$ and a target class representation $w_y$. We use the open-sourced code released by the authors in our experiment; 4) The loss function discovered via the teacher in L2T-DLF. The teacher models are optimized with Adam [26] and the detailed setting is in Appendix.

The classification results on MNIST, CIFAR-10 and CIFAR-100 are respectively shown in Table 1 and 2. As can be observed, on all the three tasks, the dynamic loss functions outputted via teacher model help to cultivate better student model. For example, the teacher model helps WRN to achieve 3.42% classification error rate on CIFAR-10, which is on par with the result discovered via automatic architecture search (e.g., 3.41% of NASNet [57]). Furthermore, our dynamic loss functions for DenseNet on CIFAR-10 reduces the error rate of DenseNet-BC (*k*=40) from 3.54% to 3.08%, where the gain is a non-trival margin.

### 4.1.1 Teacher Optimization

In Fig. 3, we provide the dev measure performance along with the teacher model optimization in MNIST experiment, the student model is LeNet. It can be observed that the dev measure is increasing along with the teacher model optimizing, and finally converges to a high score.

### 4.1.2 Analysis Towards the Loss Functions

To better understand the loss functions outputted via teacher model, we visualize the coefficients of some loss functions outputted by teacher model for training ResNet-8 in CIFAR-100 classification task. Specifically, note that the loss function $l_{\Phi_t}(p_\omega, y) = -\sigma(\vec{y}'\Phi_t \log p_w)$ essentially characterizes the *correlations among different classes* via the coefficients $\Phi_t$. Positive $\Phi_t(i, j)$ value means positive correlation between class $i$ and $j$ that their probabilities should be jointly maximized whereas negative value imposes negative correlation and higher discrimination between the two classes $i$ and $j$. We choose two classes in CIFAR-100: the *Otter* and *Baby* as class $i$ and for each of them pick several representative classes as class $j$. The corresponding $\Phi_t(i, j)$ values are visualized in Fig. 4, with $t = 20, 40, 60$ denoting the coefficients outputted via teacher model at $t$-th epoch of student model training. As can be observed, at the initial phase of training student model ($t = 20$), the teacher model chooses to enhance the correlation between two similar classes, e.g, *Otter* and *Dolphin*, *Baby* and *Boy*, for the sake of speeding up training. Comparatively, when the student model is powerful enough ($t = 60$), the teacher model will force it to perform better in discriminating two similar classes, as indicated via the more negative coefficient values $\Phi_t(i, j)$. The variation of $\Phi_t(i, j)$

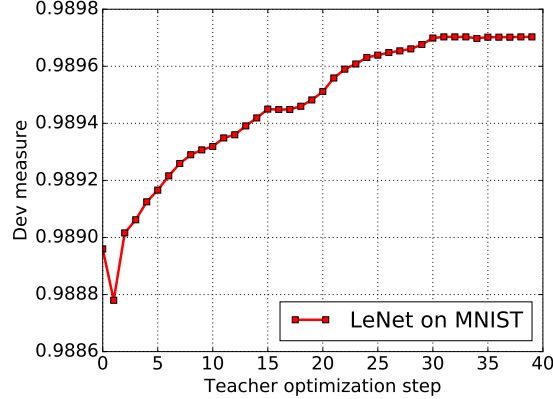

Figure 3: Measure score on the MNIST dev set along the teacher model optimization. The student model is LeNet.

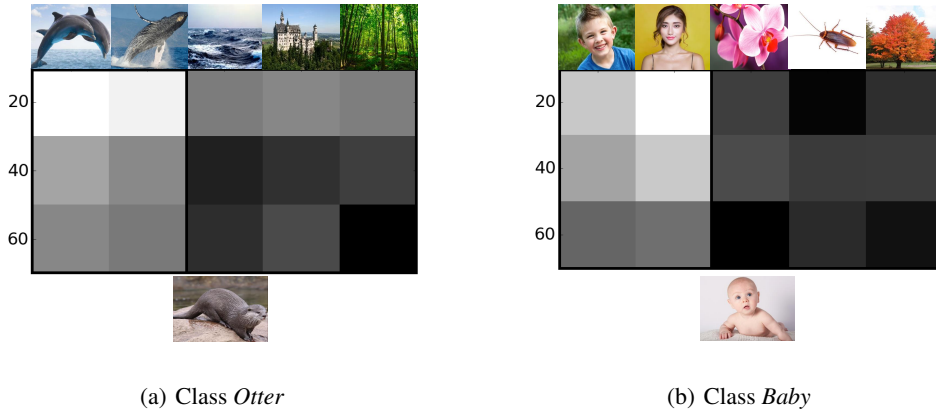

(a) Class *Otter*  (b) Class *Baby*

Figure 4: Coefficient matrix $\Phi_t$ outputted via teacher model. The y-axis $(20, 40, 60)$ corresponds to the different epochs of the student model training. Darker color means the coefficients value are more negative while shallower color means more positive. In each figure, the leftmost two columns denote similar classes and the rightmost three columns represent dissimilar classes.

values w.r.t. $t$ well demonstrates the teacher model captures the status of student model in outputting correspondingly appropriate loss functions.

## 4.2 Neural Machine Translation

In the task of *neural machine translation* (NMT), the evaluation measure $m(\hat{y}, y)$ is typically the BLEU score [41] between the translated sentence $\hat{y}$ and ground-truth reference $y$. The student model $f_\omega$ is a neural network performing sequence-to-sequence generation based on models including RNN [47], CNN [16] and self-attention network [51]. The decoding process of $f_\omega$ is typically *autoregressive*, in that $f_\omega$ factorizes the translation probability as $p_\omega(y|x) = \prod_{r=1}^{|y|} p_\omega(y_r|x, y_{<r})$. Here $p_\omega(\cdot|x, y_{<r})$ is the distribution on target vocabulary $\mathcal{V}$ at the $r$-th position, taking the source side sentence $x$ and the previous words $y_{<r}$ as inputs. Similar to the classification task, the loss function generalizing cross entropy loss is $l_\Phi = -\sum_{r=1}^{|y|} \sigma(\vec{y_r'} diag(\Phi) \log p_\omega(\cdot|x, y_{<r}))$, where $\Phi \in \mathcal{R}^{|\mathcal{V}|}$ is the coefficients of the loss function and $diag(\Phi)$ denotes the diagnoal matrix with $\Phi$ as its diagonal elements. Here we set the interaction matrix as diagonal mainly for the sake of computational efficiency, since the target vocabulary size $|\mathcal{V}|$ is usally very large (e.g., $30k$). The teacher model then outputs $\Phi_t$ at timestep $t$ taking $s_t$ as input: $\Phi_t = \mu_\theta(s_t) = W softmax(V s_t)$, where teacher model parameter $\theta = \{W \in \mathcal{R}^{|\mathcal{V}| \times N}, V \in \mathcal{R}^{N \times |s_t|}\}$. We set $N = 5$ and for the state vector $s_t$, it is the

Table 3: The translation results (BLEU score) on IWSLT-14 German-English task.

| Student Model/ Loss | Cross Entropy [52] | RL [42] | AC [3] | Softmax-Margin [12] | L2T-DLF |
|---|---|---|---|---|---|
| *LSTM-1* | 27.28 | 27.53 | 27.75 | 28.12 | **29.52** |
| *LSTM-2* | 30.86 | 31.03 | 31.21 | 31.22 | **31.75** |
| *Transformer* | 34.01 | 34.32 | 34.34 | 34.46 | **34.80** |

same with that in classification except: 1) the training/dev set accuracy is now replaced with BLEU scores; 2) the last ten features in $s_t$ for classification are ignored, leading to $|s_t| = 3$.

We choose a widely used benchmark dataset in NMT literature [42, 54, 53], released in IWSLT-14 German-English evaluation campaign [9], as the test-bed for different loss functions. The student model $f_\omega$ for this task is based on LSTM with attention [4]. For the sake of fair comparison with previous works [3, 42], we use single layer LSTM model as $f_\omega$ and name it as *LSTM-1*. To further verify the effectiveness of L2T-DLF, we use a deeper translation model stacking two LSTM layers as $f_\omega$. We denote such stronger student model as *LSTM-2*. Furthermore, we also evaluate our L2T-DLF on the Transformer [51] network. The Transformer architecture is based on the self-attention mechanism [33], and it achieves superior performance on several NMT tasks. Both LSTM/Transformer student models are trained with simple SGD. In Appendix we provide the details of the LSTM/Transformer student models and the training settings of student/teacher models.

The loss functions we leverage to train student models include: 1) Cross entropy loss $L_{ce}$ to perform maximum likelihood estimation (MLE) for training LSTM/Transformer model with teacher forcing [52]; 2) The reinforcement learning (RL) loss $L_{rl}$, a.k.a, sequence level training [42] or minimum risk training [45], targets at directly optimizing the BLEU scores for NMT models. A typical RL loss is $L_{rl}(p_\omega(x), y) = -\sum_{y^* \in \mathcal{Y}} \log p_\omega(y^*|x)(BLEU(y^*, y) - b)$, where $b$ is the reward baseline and $\mathcal{Y}$ is the candidate subset; 3) The loss specified via actor-critic (AC) algorithm $L_{ac}$ [3], which approximates the BLEU score via a critic network; 4) The softmax-margin loss, which is empirically shown to be the most effective structural prediction loss for NMT [12]; 5) The loss function discovered via our L2T-DLF.

We report the experimental results in Table 3. From the table, we can clearly observe the dynamic loss functions outputted via our teacher model can guide the student model to have superior performance compared with other specially designed loss functions. Specifically, with a shallow student model *LSTM-1*, we improve the BLEU score by more than 2.0 points compared with predefined cross-entropy loss. In addition, our *LSTM-2* student model achieves 31.75 BLEU score and it surpasses previously reported best result 30.08 by [25] on IWSLT-14 German-English achieved via RNN/LSTM models. With a much stronger *Transformer* student model, we also improve the model performance from BLEU score 34.01 to 34.80. The above results clearly demonstrate the effectiveness of our L2T-DLF approach.

## 5   Conclusion

In contrast to expert designed and fixed loss functions in conventional machine learning systems, we in this paper study how to learn dynamic loss functions so as to better teach a student machine learning model. Since loss functions provided by the teacher model dynamically change with respect to the growth of the student model and the teacher model is trained through end-to-end optimization, the quality of the student model gets improved significantly, as shown in our experiments. We hope our work will stimulate and inspire the research community to automatically discover loss functions better than expert designed ones. As to future work, we would like to conduct empirical verification on tasks with more powerful student models and larger datasets. We are also interested in trying more complicated teacher models such as deeper neural networks.

## Acknowledgments

This work was partially supported by the NSFC 61573387. We thank all the anonymous reviewers for their constructive feedbacks.

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
