[Supplementary Material]

# Supplementary Materials for the Paper "Learning to Teach with Dynamic Loss Functions"

[1]**Lijun Wu**[†,*] [2]**Fei Tian,**[†] [2]**Yingce Xia,** [3]**Yang Fan**[⋆],
[2]**Tao Qin,** [1]**Jianhuang Lai,** [2]**Tie-Yan Liu**
[1]Sun Yat-sen University, Guangzhou, China    [2]Microsoft Research, Beijing, China
[3]University of Science and Technology of China, Hefei, China
[1]wulijun3@mail2.sysu.edu.cn,  stsljh@mail.sysu.edu.cn
[2]{fetia, yingce.xia, taoqin, tie-yan.liu}@microsoft.com,  [3]fyabc@mail.ustc.edu.cn

## 1  Derivations For the Updating Rules of Teacher Model Parameters

We provide derivations of Eqn. (4) and (5) in the original paper. The starting point is Eqn.(1):

$$\omega_{t+1} = \omega_t - \eta_t \frac{\partial L_{\Phi_t}(f_{w_t}, D_{train}^t)}{\partial \omega_t} = \omega_t - \eta_t \frac{\partial L_{\mu_\theta(s_t)}(f_{w_t}, D_{train}^t)}{\partial \omega_t}. \tag{1}$$

Then we have:

$$
\begin{aligned}
d\omega_t = \frac{\partial \tilde{\mathcal{M}}(f_{\omega_T}, D_{dev})}{\partial \omega_t} =& (\frac{\partial \omega_{t+1}}{\partial \omega_t})' \frac{\partial \tilde{\mathcal{M}}(f_{\omega_T}, D_{dev})}{\partial \omega_{t+1}} \\
=& (I - \eta_t \frac{\partial^2 L_{\mu_\theta(s_t)}(f_{\omega_t}, D_{train}^t)}{\partial \omega_t^2})' d\omega_{t+1} \\
=& d\omega_{t+1} - \eta_t \frac{\partial^2 L_{\mu_\theta(s_t)}(f_{\omega_t}, D_{train}^t)}{\partial \omega_t^2} d\omega_{t+1}.
\end{aligned}
\tag{2}
$$

The last equation in Eqn. (2) leverages the symmetry of Hessian matrix: for a function $g(x, y)$, $\frac{\partial^2 g}{\partial x \partial y} = \frac{\partial^2 g}{\partial y \partial x}$.

We further have the gradient of $\theta$ only incurred at timestep $t$ (i.e., via Eqn.(1)), denoted as $d\theta|_t$, is:

$$
\begin{aligned}
d\theta|_t = \frac{\partial \tilde{\mathcal{M}}(f_{\omega_T}, D_{dev})}{\partial \theta}|_t =& (\frac{\partial \omega_{t+1}}{\partial \theta}|_t)' \frac{\partial \tilde{\mathcal{M}}(f_{\omega_T}, D_{dev})}{\partial \omega_{t+1}} \\
=& -\eta_t (\frac{\partial^2 L_{\mu_\theta(s_t)}(f_{\omega_t}, D_{train}^t)}{\partial \omega_t \partial \theta})' d\omega_{t+1} \\
=& -\eta_t \frac{\partial^2 L_{\mu_\theta(s_t)}(f_{\omega_t}, D_{train}^t)}{\partial \theta \partial \omega_t} d\omega_{t+1},
\end{aligned}
\tag{3}
$$

where $\frac{\partial \omega_{t+1}}{\partial \theta}|_t$ represents the effect of $\theta$ to the value of $\omega_{t+1}$ happened only at timestep $t$, but not related with the effect to the value of $\omega_t$. Therefore we equivalently have $\frac{\partial \omega_t}{\partial \theta} = 0$ in calculating $\frac{\partial \omega_{t+1}}{\partial \theta}|_t$. The last equation in Eqn. (3) again leverages the symmetry of Hessian matrix.

By observing $d\theta = \sum_{t=0}^{T-1} d\theta|_t$, we obtain the recursive way to update $d\theta$ at timestep $t$ as in Eqn.(5) of the main paper:

$$d\theta = d\theta + d\theta|_t = d\theta - \eta_t \frac{\partial^2 L_{\mu_\theta(s_t)}(f_{\omega_t}, D_{train}^t)}{\partial\theta\partial\omega_t} d\omega_{t+1}. \tag{4}$$

## 2 Experiment Details

The details of network structures for the student models, the dataset used for neural machine translation, the training procedure for student and teacher models are provided here.

### 2.1 MNIST

For MNIST dataset, we choose the simple multi-layer perceptron (MLP) and vanilla convolutional neural network (CNN) based LeNet architecture as our student models.

The MLP contains only one single hidden layer with hidden size $500$, and the logistic regression output layer with size $10$. The input MNIST training sample is a flattened vector with size $28 \times 28$. The model is trained with mini-batch size $20$, momentum SGD [9] is adopted with learning rate $0.01$ and momentum $0.9$ in straining the student model.

The LeNet [8] model contains two (convolution + max-pooling) layers with kernel size $5 \times 5$ and filter number $20, 50$ respectively, followed by one MLP with hiden size $500$. The model is trained with mini-batch size $500$ and the learning rate for momentum SGD update is $0.01$, the momentum is $0.9$.

### 2.2 CIFAR-10/CIFAR-100

For CIFAR-10 and CIFAR-100, we use the advanced CNN architecture ResNet [4] with different number of layers, and also the Wide-ResNet [13], DenseNet [6] which has superior performance.

We use the original and typical setting for the ResNet architecture. The inputs for the network are $32 \times 32$ images, with the per-pixel mean subtracted. The first layer is $3 \times 3$ convolutions, and then stack of $6n$ layers with $3 \times 3$ convolutions on the feature maps of sizes $\{32, 16, 8\}$. The numbers of filters are $\{16, 32, 64\}$ respectively. The subsampling is performed after convolutions with a stride of $2$. The network ends with a global average pooling layer, a 10-way (for CIFAR-10) or 100-way (for CIFAR-100) fully-connected layer and softmax layer. There are totally $6n + 2$ stacked weighted layers. Identity shortcuts are connected to the pairs of $3 \times 3$ layers. We vary the $n = \{1, 3, 5\}$, leading to $\{8, 20, 32\}$-layer networks to evaluate our algorithm. The momentum optimizer with learning rate $0.1$ and momentum $0.9$ is conducted to update the student model, the learning rate is divided by $10$ after $40$ and $60$ epochs. The mini-batch size is $128$ in training. For data augmentation we do horizontal flips and take random crops from image padded by $4$ pixels on each side, filling missing pixels with reflections of original image.

For CIFAR-10 dataset, we further adopt Wide-ResNet (WRN) and DenseNet as our student model. The WRN decreases the depth and increases witdth of ResNet. The specific configuration is WRN-40-10 setting, a ResNet with $40$ convolutional layers and a widening factor $10$ (the number of filters are $10$ times wider than the original ResNet, which is $\{160, 320, 640\}$). Other details are same as ResNet setting. For the DenseNet, the configuration is same as in [6], with bottleneck layers and compression module, named as DenseNet-BC. Specifically, the layer number $L$ is $190$ and the growth rate $k$ is $40$.

### 2.3 IWSLT-14 German-English NMT

For neural machine translation (NMT) experiment, the IWSLT-14 German-English [3] dataset we choose is a well-acknowledged benchmark in NMT literature. The training/dev/test dataset respectively contains roughly $153k/7k/7k$ sentence pairs. We process the German and English sentences to be $25k$ sub-word units by byte-pair-encoding (BPE) [10] approach. The student model we used is based on LSTM [5] with attention mechanism [2] and the Transformer [12] network based on self-attention. The embedding size and hidden state size are both set as $256$. *LSTM-1* contains only

one LSTM layer while *LSTM-2* has two LSTM hidden layers. Both student models are trained with simple SGD with learning rate $0.1$, the mini-batch size is $32$. The configuration for *Transformer* is the $transformer\_small$ setting with 6 layers of the encoder and decoder. To speed up the training process, as commonly done in previous works [1, 11], we pretrain our student models for several epochs as warm-start models, and the training/dev set BLEU scores are computed based on the greedy searched translation results.

## 2.4 Teacher Optimization

For all experiments, the teacher models are optimized by Adam [7] with $\alpha = 0.0001, \beta_1 = 0.9, \beta_2 = 0.999$ and $\epsilon = 10^{-8}$. The teacher models are optimized with $60$, $100$ and $50$ steps (i.e., the number of teacher optimization steps in Algorithm 1 of the paper) for MNIST, CIFAR-10/CIFAR-100 and German-English translation tasks respectively.

## Footnotes

*The work was done when the first and fourth authors were interns at Microsoft Research Asia.

†The first two authors contribute equally to this work.