[Reviews · NeurIPS 2018]

Reviewer 1



The paper studies the framework of teaching a loss function to a machine learning algorithm (the student model). Inspired from ideas of machine teaching and recent work of “learning to teach” [Fan et al. 18], the paper proposes L2T-DLF framework where a teacher model is jointly trained with a student model. Here, the teacher’s goal is to learn a better policy of how to generate dynamic loss functions for the student by accounting for the current state of the student (e.g., training iteration, training error, test error). As shown in Algorithm#1, the teacher/student interaction happens in episodes: (1) the teacher’s parameter \theta is fixed in a given episode, (2) the student model is trained end-to-end, and (3) then \theta is updated. In Section 3.3, the paper proposes a gradient-based method to update the parameter of the teacher model. Extensive experiments are performed on two different tasks to demonstrate the effectiveness of the proposed framework. Please see a few comments below: (i) In my opinion, the paper is trying to overemphasize the connections with machine teaching literature and real-life classroom teaching in the Abstract and Introduction. The technical results of this paper have a very weak connection with machine/real-life teaching. In the proposed L2T-DLF framework, the teacher and student models are just two components of an end-to-end learning system. For instance, in Algorithm #1, the teacher model essentially tunes its policy by training the same student model again and again. This is fine for a learning system but somewhat disconnected from real-life teaching scenarios or machine teaching literature. (ii) Authors should expand the discussion of how their work technically differs from learning to teach framework [Fan et al. 18] as well as techniques of curriculum learning / self-paced learning. For an iterative learner such as those studied in [Liu et al . 2017, reference 30], the teacher’s choice of the training instance $x_t$ at time $t$ can equivalently be seen as choosing an appropriate loss function $l_t$. (iii) A few points to improve the presentation: - Line 121: Perhaps use $N$ for the number of data points instead of $M$. You are already using $m$ for the task-specific objective and \mathcal{M} for a score. - Mixing terms “dev”, “development”, and “test”. (iv) Appendix: I believe Section 2.4 “Teacher Optimization” and Figure#1 are important, and should be moved to the main paper.

Reviewer 2



Let me explain why I think the algorithm is the same as Andrychowicz et.al., with "the new interpretation of such algorithm and the parametrization in terms of machine teaching scenario" as in my review. Following the same notations used in this paper, let's check the update rule in L2T-DLF: w_{t+1} = w_t - \eta_t g(w_t; \theta) where g(w_t;\theta) = \nabla_\theta L(w; \theta). Conduct this update T times, we obtain w_T which can be understood as a RNN with \theta as the parameters. Plug this into the Eq (2), \theta is then optimized by the (stochastic) gradient descent. Essentially, the L2T-DLF is learning an update rule in the optimizer, which is the same as Andrychowicz et.al.. There might be two differences: 1, in L2T-DLF, the update rule is derived from loss function perspective, and in Andrychowicz et.al., they directly parametrized the update rule as a RNN. 2, the objectives of the optimizer (as clarified by the authors in the reply) Andrychowicz et.al. use LSTM and working on multiple datasets, while L2T-DLF use a traditional RNN and working on training and development dataset, which makes these two different. However, these are the parametrization and loss function, which I think should be known as "model", while I am talking about the algorithm to obtain the learned optimizer. I think these connections should be discussed in the paper. Regarding the machine teaching protocol, the author assumes the teacher not only know the students updates rule and stepsize, but also can *modify* these in students, i.e., the students will use the first-order of the teacher's output for future learning. This is different from the existing literature [30]. The teaching protocol is important in machine teaching. With different ability of the teacher, the teaching efficiency will be different. For an extreme example, if the teacher can directly manipulate the student's model, then, no sample will be needed. As a machine teaching paper, the teaching protocol should be clearly discussed. In sum, I think at least this interpretation is interesting. However, the authors should discuss the connections to the existing work, especially Andrychowicz et.al., and the teaching protocol. ============================================= In this paper, the authors raise an interesting teaching scheme, i.e., the teacher adapts the loss functions for the students, which leads to the changes in the update rule of the students. By parametrization of the loss functions, with the reverse-model differentiation trick, the loss function of teacher and the model of the student can be updated together. The authors evaluate the proposed algorithm on several practical applications, including image classification and neural machine translation, and demonstrate the benefits of the proposed method. However, there are several issues in current version need to be addressed. 1, The proposed algorithm is **exactly the same** algorithm proposed in [1]. The novelty part is the new interpretation of such algorithm and the parametrization in terms of machine teaching scenario. However, such important reference and other related work [2, 3] has never be appropriately cited and discussed. Please add the discussion to the related work [1] and its extensions [3] and emphasize your contribution comparing to [1,2,3]. 2, The machine teaching scheme is not clearly explained. Specifically, based on the algorithm proposed in the paper, it seems the teacher can access to the zero- and first-order of the model of the students, and moreover, the students can access to the first-order of the teachers loss function. The differences communication scheme between the proposed teaching scenario and the existing teaching scenario should be discussed clearly. 3, The reverse-model differentiation has a well-known drawback. The memory cost are increasing with the steps of the unrolling of the first-order updates, i.e., T in the Algorithm 1. Such drawback limits the RMD usage in practice and should be discussed in the paper. In sum, I think the interpretation of the learning to optimization idea in terms of machine teaching is novel and inspiring. However, the existing work should be correctly acknowledged, and the drawback should be explicitly discussed. [1] Marcin Andrychowicz, Misha Denil, Sergio Gomez, Matthew W Hoffman, David Pfau, Tom Schaul, and Nando de Freitas. Learning to learn by gradient descent by gradient descent. arXiv preprint arXiv:1606.04474, 2016. [2] Li, Ke and Malik, Jitendra. Learning to optimize. International Conference on Learning Representations (ICLR), 2017. [3] Chen, Y., M. W. Hoffman, S. G. Colmenarejo, M. Denil, T. P. Lillicrap, M. Botvinick, and N. Freitas (2017). Learning to learn without gradient descent by gradient descent. In: International Conference on Machine Learning, pp. 748–756

Reviewer 3



** Post Response ** A thorough discussion regarding the connections to meta-learning techniques [3-7] should be added to the paper (and would significantly strengthen the paper, as it would make it more accessible and more likely to have impact within the meta-learning community). I hope that the authors include this discussion in the final version, as they say they will in the author response. The new experiments and results address my other primary concerns, demonstrating that the method performs very strongly on a number of challenging, large-scale problems and generalizes to new datasets and architectures. I am raising my score from a 7 to an 8. ** Original Review ** This paper aims to learn a differentiable loss function that leads to good performance on the "true" objective for a particular task (e.g. accuracy or BLEU score). In particular, the loss function is optimized with gradient descent with reverse mode autodiff w.r.t. the true objective on held-out data. The experiments include comparisons to prior work on MNIST, CIFAR-10, CIFAR-100, and De-En NMT, showing a small improvement. Pros: The idea is interesting and motivated well, from the perspective of a teacher adapting how it teaches a student. The experiments include a number of different datasets, many different model architectures, and comparisons to multiple previous methods. Cons: The improvement over prior work seems to be relatively small (but it is consistent across tasks). There are important aspects of the experimental set-up that are unclear (details below). Finally, the proposed approach seems to be very computationally intensive, since it requires training the teacher on the task and then training the student using the teacher. After being trained, can the teacher generalize to new tasks? Method: As mentioned above, the method is nicely motivated, from the perspective of a teacher. But, the method in its current form seems somewhat impractical, as it requires training a teacher from scratch for a particular task and then training the student. It seems like it would be much more practical to train the teacher across multiple tasks and measure generalization to new tasks. Can the teacher generalize to new settings? Regarding reproducibility, will code be released? Experiments: How does the proposed approach compare to reward-augmented max likelihood [1] and distillation [2]? [1] considered the same problem of optimizing for a non-differentiable reward function. Model distillation [2] also has a similar flavor of a teacher/student where the teacher is an ensemble of models and the student is a single model. One crucial piece of information that is missing from the experiments is -- what data is used for the development set, and how was this data used for the comparisons? The concern is that, perhaps the proposed method is seeing a benefit because it is using the validation set in a more direct way, rather than simply being used to find good hyperparameters. If the dev set is only used by the prior methods for choosing hyperparameters, how do these prior methods perform when training with both the training and development data (after finding good hyperparameters by holding out the dev set)? Additionally, when experimenting with multiple student architectures, was a single teacher trained across all student architectures? Or was it trained individually for each architecture? Or was the teacher trained on one student architecture and evaluated on another? This bit of information doesn't seem to be stated in the paper. Showing generalization to new student architectures would be nice. Related Work: It would be nice (but not crucial) to discuss the relationship with works on learning to optimize [3,4]. These works are related in that they learn a meta-learner/teacher that learns how to update learner/student. Unlike this work, that learns a loss function, these prior works learn to directly output the weight update to be applied to learner/student. There is also prior work in learning-to-learn that learns critics or loss functions across multiple tasks [5,6,7]. Of course, one big difference with this work and these prior works is that these prior works typically train over multiple tasks or multiple architectures, evaluating generalization to new settings. This paper also seems to be related to automatic curriculum generation, e.g. [8]. Minor comments: - The notation for y in Section 4.1 is inconsistent. What is the difference between y, y with an arrow, and y' with an arrow? [1] Reward Augmented Maximum Likelihood for Neural Structured Prediction https://arxiv.org/abs/1609.00150 [2] Distilling the Knowledge of a Neural Network https://arxiv.org/abs/1503.02531 [3] Learning to Learn by Gradient Descent by Gradient Descent https://arxiv.org/abs/1606.04474 [4] Learning to Optimize https://arxiv.org/abs/1606.01885 [5] Meta-critic Networks https://arxiv.org/abs/1706.09529 [6] Domain Adaptive Meta-Learning https://arxiv.org/abs/1802.01557 [7] Evolved Policy Gradients https://arxiv.org/abs/1802.04821 [8] Automatic Curricula via Asymmetric Self-Play https://arxiv.org/abs/1703.05407